# A Hybrid Brain–Computer Interface for Real-Life Meal-Assist Robot Control

**DOI:** 10.3390/s21134578

**Published:** 2021-07-04

**Authors:** Jihyeon Ha, Sangin Park, Chang-Hwan Im, Laehyun Kim

**Affiliations:** 1Center for Bionics, Korea Institute of Science and Technology, Seoul 02792, Korea; hj910410@kist.re.kr (J.H.); sipark@kist.re.kr (S.P.); 2Department of Biomedical Engineering, Hanyang University, Seoul 04763, Korea; ich@hanyang.ac.kr; 3Department of HY-KIST Bio-Convergence, Hanyang University, Seoul 04763, Korea

**Keywords:** meal-assist robot, brain–computer interface, electroencephalogram, steady-state visual evoked potential, eye-blink, electromyogram

## Abstract

Assistant devices such as meal-assist robots aid individuals with disabilities and support the elderly in performing daily activities. However, existing meal-assist robots are inconvenient to operate due to non-intuitive user interfaces, requiring additional time and effort. Thus, we developed a hybrid brain–computer interface-based meal-assist robot system following three features that can be measured using scalp electrodes for electroencephalography. The following three procedures comprise a single meal cycle. (1) Triple eye-blinks (EBs) from the prefrontal channel were treated as activation for initiating the cycle. (2) Steady-state visual evoked potentials (SSVEPs) from occipital channels were used to select the food per the user’s intention. (3) Electromyograms (EMGs) were recorded from temporal channels as the users chewed the food to mark the end of a cycle and indicate readiness for starting the following meal. The accuracy, information transfer rate, and false positive rate during experiments on five subjects were as follows: accuracy (EBs/SSVEPs/EMGs) (%): (94.67/83.33/97.33); FPR (EBs/EMGs) (times/min): (0.11/0.08); ITR (SSVEPs) (bit/min): 20.41. These results revealed the feasibility of this assistive system. The proposed system allows users to eat on their own more naturally. Furthermore, it can increase the self-esteem of disabled and elderly peeople and enhance their quality of life.

## 1. Introduction

The population of elderly and disabled individuals is increasing worldwide [1,2]. Currently, these individuals experience challenges in their lives without therapists. In particular, their self-esteem decreases if they cannot even perform basic activities such as walking and eating, which consequently increases their helplessness. Recently, robots have been actively developed for applications in healthcare [3], such as home-assist robots [4], exoskeleton robots [5], and meal-assist robots [6]. These robots have progressed from being directly controlled to completely automatic applications. However, the application of completely automatic methods may not be user friendly [7]. Thus, many researchers have conducted studies based on human–computer interactions (HCIs) [8,9]. There are various studies on healthcare robots based on augmented/virtual reality (AR/VR) [9], voice [7], and electroencephalograms (EEG) [5,6].

Non-invasive brain–computer interfaces (BCIs) are HCI methods based on non-invasive measurements such as EEG [10,11]. Several scholars have studied the daily-life application of non-invasive BCIs: assist robots [5,12], smart home [13,14,15], emotion detectors [16], and wheelchairs [17]. In particular, BCI-based assist robots are expected to be in practical application for the disabled and elderly population. Typically, the methods used for transferring human intentions include motor imagery (MI), event-related potential (ERP), and steady-state evoked potential (SSVEP) [5,13,15]. Research based on MI is advantageous for intuitively controlling the devices. However, the MI method requires considerable training, and increasing the number of classes is a challenging task. Unlike MI, ERP and SSVEP are not intuitive, i.e., they have to associate meaning to stimuli. Although the number of stimuli in ERP and SSVEP is greater than that in MI, these methods deliver more accurate results in general cases [18,19]. The SSVEP-based methods generally require no training sessions, unlike ERP [20,21]. As above, each BCI method has its own set of advantages and disadvantages.

Nevertheless, the BCI system requires an activation for its initiation, which accounts as a significant disadvantage [19]. Therefore, a system that uses only EEG can yield unnatural usage. For instance, the stimulus must be turned on at all times to utilize the ERP or SSVEP. Moreover, additional actions must be performed to activate the operation of MI. Recently, several BCI systems have been constructed by combining EEG and other signals [12,13,22]. Park et al. (2020) used eye blinks to set the initiation point of the stimulus [13]. Gao et al. (2017) combined MI and EMG to set the initiation point for stimuli [12]. Such hybrid-BCI studies have considered the practical convenience of using BCI systems.

In this study, we developed a healthcare robot, more specifically, a meal-assist robot in an EEG-based hybrid-BCI system, which utilizes only the signals obtained from the scalp electrodes. The preview of the final system is stated as follows: (1) triple eye-blinks (EBs) from the prefrontal channel were used as an activation for initiating a single meal cycle; (2) SSVEPs from occipital channels were used to select the food per the user’s intention; and (3) EMGs were recorded from temporal channels as the users chewed their food, indicating the end of the cycle and the readiness to start the following meal. In order to verify the feasibility of this system, we measured the performance of algorithms by conducting experiments on five subjects. The proposed system allows users to eat their meals more naturally, unlike conventional systems that require assistance.

## 2. Materials and Methods

### 2.1. Experiment for Measuring Performance

#### 2.1.1. System Configuration

The system for experiments used two computers: one (Intel CPU i7-8565U 1.80 GHz, RAM 16 GB) for measuring the EEG, and another (Intel CPU i7-8550U 1.80 GHz, RAM 24 GB) for using E-Prime 3.0 (Psychology Software Tools, Pittsburgh, PA, USA) to control the LED and present experimental instructions. In addition, each computer communicated over the TCP/IP. Moreover, Active Two (Biosemi S. V., Amsterdam, The Netherlands) and 64 channels of pin-type active electrodes were used to measure the EEG signals. The data acquisition software was a lab streaming layer (LSL) [23] and MATLAB (2020a, Mathworks Inc., Natick, MA, USA). A data acquisition (DAQ) board (USB-6501, National Instrument Corp., Austin, TX, USA) was used as well. The discussed experimental setup and its devices are presented in Figure 1.

#### 2.1.2. Experimental Protocol

Prior to the start of the experiment, the subjects were presented with instructions for the EB, observing the indicated LED for SSVEP, and chewing gum for EMG. One cycle of an experiment is given as follows. Initially, the monitor displayed the message: “Do triple blinking” for 3 s, and the subjects conducted EB within 3 s as a fixation (+) appeared. Following, the monitor displayed a picture of a particular food on the tray for 3 s, and the subjects were required to look at the target LED for 5 s as the fixation (+) appeared. Finally, the monitor displayed the message: “Chew gum” for 3 s, and the subjects chewed gum for 5 s when the fixation (+) appeared. According to the five types of SSVEP stimuli, the five distinct cycles were conducted in a random order. Five cycles were comprised under a single trial, and the subjects had a rest period of 10 s between trials. In total, six trials (30 cycles) of experiments were conducted. Moreover, the SSVEP stimuli were identical to those of the real-time system (rice 1, side dish 4), and the five stimuli were selected as 7.4, 8.43, 9.8, 11.7, and 13.7 Hz. The experimental paradigm is illustrated in Figure 2. The experimental video (single cycle) is enclosed with the current article as a Appendix A, which can be found at https://youtu.be/CfVYz_cMFto as well (accessed on 24 May 2021).

#### 2.1.3. Participants

We recruited five healthy males (age: 29.40 ± 3.13) to participate in the experiment. None of the subjects have ever participated in an LED-based SSVEP experiment. After participating in the experiment, we rewarded the participants with monetary remuneration. This experimental study was approved and reviewed by the Institutional Review Board (approval number: 2019-032) of the Korea Institute of Science and Technology (KIST).

#### 2.1.4. Data Processing and Analysis

Figure 3 shows the procedures for the processing and detection of EBs, SSVEPs, and EMGs using the sample data (subject 2). The sampling rate for all the signals was 512 Hz. We used the signal obtained from the electrodes (FPz) to acquire the eye blinks. Some previous studies used data obtained from FPz for eye-blinks [22,24,25]. The data were smoothed and the DC components were removed using an elliptic infinite impulse response (IIR) bandpass filter (high cut-off frequency: 5 Hz, low cut-off frequency: 0.5 Hz) in MATLAB [26]. We found the peaks using the toolbox of MATLAB (“findpeaks”). The detected peaks were screened to avoid a mistake-blink (very fast blinking) on the basis of: (1) were the numbers of the condition peak value > threshold (mean amplitude of each subject/2) three?; (2) 0.3 s < peak-to-peak interval < 0.8 s; (3) if there was no EB, we used continuous wavelet-transform and found peaks; and (4) we repeated steps (1) and (2). Thus, an EB was reported upon detection, if these conditions were satisfied.

Moreover, the channels in the occipital region (O1, Oz, and O2) were used to measure the SSVEP, and the data were re-referenced by subtracting the Cz signal. In addition, an elliptic IIR bandpass filter (high cut-off frequency: 54 Hz, low cut-off frequency: 2 Hz) was used (frequency of line noise in South Korea: 60 Hz). The SSVEP was detected using the extension to the multivariate synchronization index (EMSI) [21], as several studies have applied this method for SSVEP-based BCI systems [4,13].

The channels in the temporal region (T7 and T8) were used to measure the EMG, and an elliptic IIR highpass filter (low cut frequency: 0.5 Hz) was used to remove the DC components. (1) The moving average was evaluated after considering the absolute value; (2) the differentiation was performed; (3) the moving average was determined after accounting for the absolute value; and (4) counting the number of samples that satisfied the condition processed data value > threshold (50% of median value of each subject). If the counted samples (instances) were greater than 1024 samples (2 s), we reported that EMG was detected.

The performance of the developed system was verified by calculating the accuracy (unit, %) of the eye-blink/SSVEP/EMG, information transfer rate (ITR) (unit, bit/min) of SSVEP, and false positive rate (FPR) (unit, times/min) during the entire experimental period (10.5 min). The ITR is the standard method for measuring the performance of communication in control systems, especially for BCI-based systems. The ITR denotes the amount of information transferred per time. Detailed descriptions and equations for ITR are stated in Appendix B as well.

### 2.2. Real-Time System

#### 2.2.1. System Configuration

The proposed system used two computers: one (Intel CPU i7-8565U 1.80 GHz, RAM 16 GB) for measuring EEG and another (Intel CPU i7-7700HQ 2.80 GHz, RAM 16GB) to control the LED and meal-assist robot, wherein each computer communicated over TCP/IP. The EEG measurement device was identical to that used in the experiment for measuring performance. However, in the real-time system, 64 channels were not used. A total of seven flat-type active electrodes (FPz, T7, O1, Oz, O2, CMS, and DRL) were used. The meal-assist robot, “Caremeal,” used in this study was manufactured by NT robot (2004; Seoul, Korea) [27]. Caremeal comprised a spoon (2-axis motor) and a grab (5-axis motor) arm. In addition, the DAQ board for LED control was identical to that in the experiment for measuring performance. For real-time data acquisition, we used the OpenVibe [28] acquisition server with LSL, and the acquired data were analyzed using the OpenVibe designer with MATLAB. Figure 4 illustrates the devices and settings of the real-time system.

#### 2.2.2. Data Acquisition and Processing

In the real-time system, the sampling rate of all channels, the high/low cutoff frequency of the bandpass filter, and the epoching time were the same for efficient processing. The sampling rate was 128 Hz, and an elliptic IIR bandpass filter (0.5–55 Hz) was used. The signal window was of 4 s, and the sliding period was 1 s. Algorithms detecting EBs, SSVEPs, and EMGs were identical to that in the experiment for measuring performance. Figure 5 depicts the data acquisition and processing procedure in a real-time system.

## 3. Results

### 3.1. Experimental Result

First, we analyzed the SSVEP based on the following conditions: (1) one channel: Oz, epoch period: 3 s (average accuracy (%): 71.33, average ITR (bit/min): 18.46); (2) three channels: O1/Oz/O2, epoch period: 3 s (average accuracy (%): 70.67, average ITR (bit/min): 17.47); (3) one channel: Oz, epoch period: 4 s (average accuracy (%): 77.33, average ITR (bit/min): 17.26); and (4) three channels: O1/Oz/O2, epoch period: 4 s (average accuracy (%): 83.33, average ITR (bit/min): 20.41). The detailed results are presented in Table 1. The SSVEP results of (4) are depicted in Figure 6, which is the most suitable and accurate among the EB and EMG results. The average accuracy (%) of the EB was 94.67 and that of the EMG was 97.33. The averages of the FPRs during the complete experimental period were 0.11 times/min for EB and 0.09 times/min for EMG. The detailed results are presented in Table 2. We made the preprocessed data (five samples) and classification codes (EB, SSVEP, and EMG) public at https://github.com/devhaji/HBCI_MAR_Sample (accessed on 29 June 2021).

### 3.2. Simulation of the Proposed System

A single cycle of meal can be stated as follows: (1) EB was detected; (2) five LED flickers appeared and the SSVEP response was induced; (3) the grabbing arm of the meal-assist robot moved to select the food based on the SSVEP results; (4) the grabbing arm transferred food onto the spoon, and the spoon arm fed the subject; (5) the subject chewed the food, and the EMG was detected. The meal was considered to have ended after 10 s if the EMG was not detected; and (6) the spoon arm returned to the original position, and the system was ready for the following meal or ended. EBs can be detected when the system is ready for the following meal. Thus, EBs were not detected during meals. Figure 7 illustrates a schematic of the real-time system. The demo video (single meal cycle) is enclosed with this article as a Appendix A and can be found at https://youtu.be/CfVYz_cMFto as well (accessed on 24 May 2021).

## 4. Discussion

This study implemented a hybrid BCI-based meal-assist robot, wherein the accuracy of the algorithm was examined, based on an experiment to verify the feasibility of a real-time practical system. As the purpose of this study is to achieve a real-time application, the use of the fewest possible channels was recommended with minimal epoch. However, the SSVEP accuracy was poor in the case of using 3 s of data. Interestingly, the one channel and 3 s of data revealed better results than the three channels and 3 s of data. This was due to the synchronization value that was the output of the EMSI algorithm between covariance data (three channels), and target frequencies were smaller than that of using the one channel. Zhang et al. (2017) reported that the accuracy of EMSI increased as the time window increased [21]. In our study, the accuracy using 4 s of data was higher than 3 s of data, and the ITR was higher than 20 bit/min. Finally, we decided that the suitable time window is 4 s, and the number of channels is three, in consideration of maximum accuracy and usability. Therefore, we used three flat-type electrodes and 4 s for the SSVEP in the real-time system. According to the experiment, EBs and EMG exhibited exceedingly high accuracy. The same algorithms were conveniently applied in a real-time simulation as they did not require considerable computational load.

In a real-time system, stability is as equally important as accuracy. We calculated the FPR of the EB and EMG, in which extremely low average FPRs were reported. However, these FPRs were not zero. Han et al. (2020) developed a BCI toggle switch with very low FPR (0.02 times/min) and high accuracy (100%) using respiration-modulated photoplethysmography [29]. If the EBs are not detected during meals, the FPR of our system may be almost zero, as in the previous study. Likewise, almost zero FPR for EMG is possible if the EMG can be detected only during the operation of the spoon arm. Nonetheless, the descent of the spoon could be more dangerous if the EMG was incorrectly measured than to start eating when the EB was incorrectly measured. Therefore, we included a slightly stricter criterion for the real-time system. In the experiment, we decided that the “EMG was detected” if the EMG was maintained for > 2 s. In the real-time system, the criterion required the maintenance of EMG for more than 2 s, which must be detected three times. The spoon arm returned to its original position after 10 s if the EMG was not detected.

There is a similar study based on the proposed system, in which Perera et al. (2016) suggested the concept of a meal-assist robot with an SSVEP-based BCI [6,30]. However, the current study differs from the previous study on certain measures. (1) The meal-assist robot used in their research comprised a single robot arm, whereas that used in our study comprised two robot arms, because we decided that two robot arms are required to feed the Korean-style meal [27]—one as the grab arm for grabbing the food, and the other as a spoon arm for feeding. (2) Their study used three LED-based stimuli, but this study used five LEDs. Although this study reported the precise performance of all measurements in the experiment, the performance of their study could not be reported clearly. (3) Their study was for offline systems, and not for online systems, because their system was based on asynchronous BCI as the algorithm could not distinguish between the idle and control state without aid. There have been numerous prior research studies aiming at the development of highly accurate asynchronous BCI [31,32]. Therefore, several recent studies on real-life BCI [5,13], including the current study, used eye-blink as an activation mechanism. (4) The proposed system used EMG for real-life purposes to detect chewing a morsel, on which the system determined the end of the user’s a single meal cycle. Thus, the proposed system can be stabilized if the spoon arm only moves to its original position after the user finishes a single meal cycle. Therefore, Perera et al. (2016) were the first to suggest the concept of a meal-assist robot. This study focused on a hybrid BCI-based meal-assist robot for real-life applications.

This study only reported experimental results for five male subjects. Although the EB and EMG results did not vary between the sexes, a previous study has reported that the SSVEP was detected better in males [33]. Therefore, we conducted the experiment only on men, and applying the results of this study on women should involve caution. The experiments with the disabled are more important for a BCI-based meal-assist robot. In future, this system will be used to conduct research that can be applied to the disabled, and the accuracy of SSVEP and FPR of EB and EMG will be calculated to verify the real-time system. Recently, several studies of SSVEP-based BCIs for robot control reported higher accuracy and ITR [34,35,36]. We discussed the reason for the low accuracy obtained in this study in comparison to previous studies. When the subject watched the LEDs on the tray, a certain amount of light reflected on the tray owing to LED interference. However, the purpose of this system is to intuitively eat meals. The operation of this hybrid BCI-based meal-assist robot requires approximately 10 s longer if a joystick is used for a single meal cycle, considering that the joystick user has not made any mistakes and the BCI is 100% accurate. Thus, this system can aid disabled and elderly individuals if the process is natural and not uncomfortable for its intended users. However, the problem regarding the interference of the reflected light should be resolved because the user will want to eat as per his/her desire. Overall, the impact of reflection will be diminished if the position and the height of LEDs are considered. Alternatively, the AR can provide an adequate solution to this problem, which is similar to other applications [13,37,38]. As a future scope of research, we will modify previous system or alter the LED-based stimuli as AR-based stimuli.

This study was conducted at the laboratory level as a preliminary study to test the hybrid-BCI system-based meal assist robot on the disabled and elderly. For practical use, the system had the following characteristics: (1) a signal for detecting EBs (FPz), signals for detecting SSVEP (O1, Oz, O2), and signals for detecting EMG (T7, T8) from a single device were measured. (2) We used algorithms requiring no training. (3) We limited when the system can detect EBs and EMG to increase the FPR. Nevertheless, there will be additional considerations for this system to be applied in practice. One of these considerations is, like previous assistant devices, the need of a caregiver or a therapist for switching the system on/off and device-setting. In this study, the EMG was measured by chewing gum. In actual use, since EMG can appear differently depending on the food eaten [39], input parameters for thresholding may be required. The user may urgently need to stop the system, and EEG may be exposed to various artifacts. In future studies on the disabled and elderly, therefore, the preparation process for use will also be an important consideration. In addition, tests of various situations will be required to compare laboratory-level parameters with those for practical use. Functions such as strategy for artifact removal to ensure the accuracy of the system and an emergency stop to increase stability should be also considered.

## 5. Conclusions

The SSVEP-based BCI delivered high accuracy and generally did not require any training [20,21]. Therefore, the proposed hybrid BCI-based meal-assist robot was based on the SSVEP. We used eye-blink from the prefrontal area and EMG from the temporal area, which have been considered as artifacts in previous BCI systems. In particular, the proposed system only used EMG to detect chewing a morsel. Based on these signals, the user could eat their meals naturally using the meal-assist robot. Thus, the proposed system can increase the self-esteem among disabled and elderly individuals and enhance their quality of life.

## Figures and Tables

**Figure 1 sensors-21-04578-f001:**
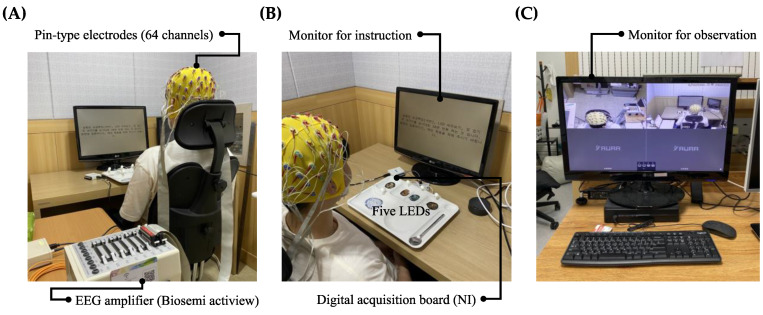
Experimental setup. (**A**) Device for EEG acquisition (EEG amplifier and 64 channels pin-type electrodes). (**B**) Monitor for instruction, a tray with five LEDs connected to digital acquisition board. (**C**) Monitor for observation.

**Figure 2 sensors-21-04578-f002:**
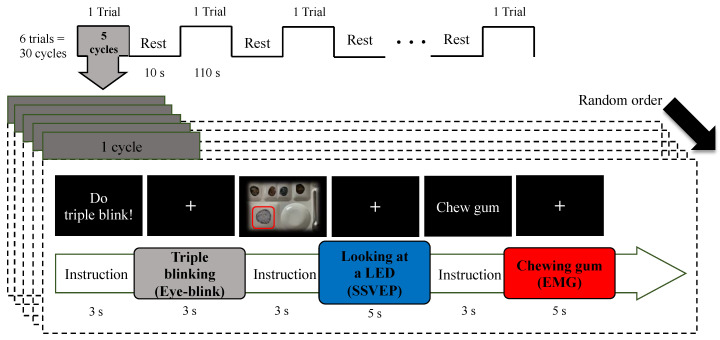
Schematic diagram of experimental protocol.

**Figure 3 sensors-21-04578-f003:**
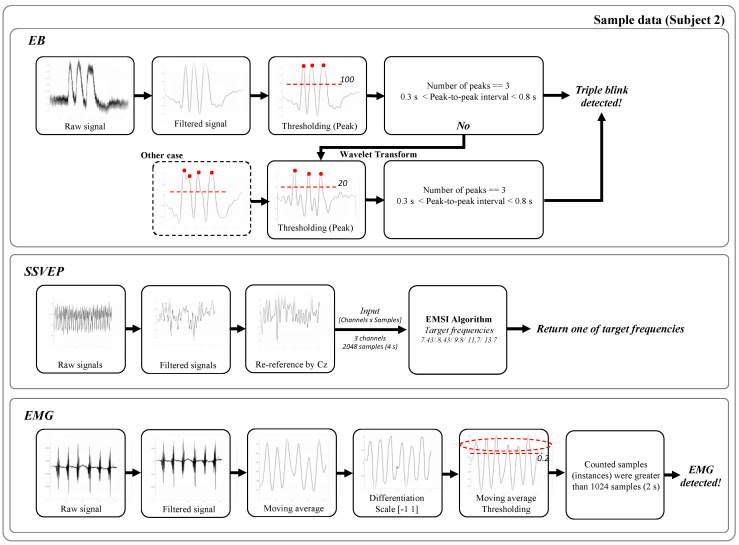
Procedure of processing and detection of triple eye-blink (EB), steady-state evoked potential (SSVEP), and electromyogram (EMG). Extension to multivariate synchronization index (EMSI).

**Figure 4 sensors-21-04578-f004:**
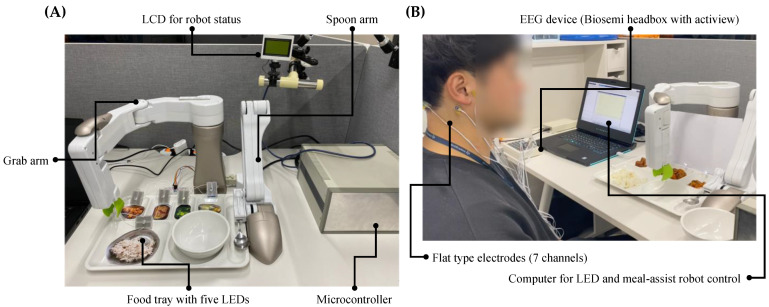
Devices for real-time system. (**A**) Components of meal-assist robot and related parts. (**B**) Devices for EEG acquisition.

**Figure 5 sensors-21-04578-f005:**
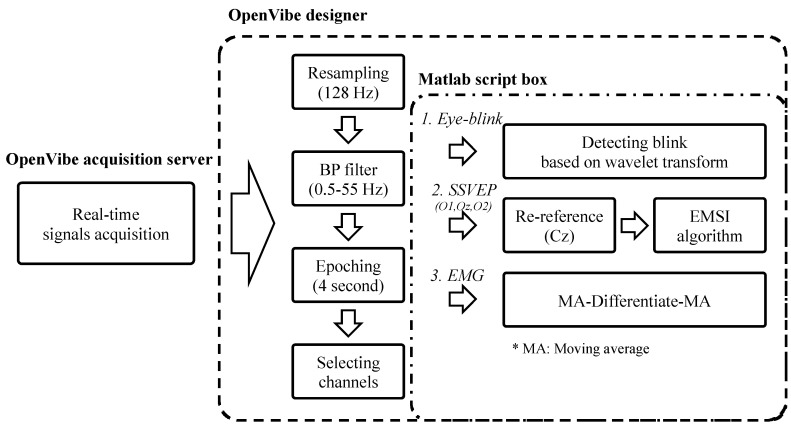
Procedure of EEG acquisition and processing in real-time system. Elliptic infinite impulse response filter bandpass filter, BP filter; steady-state evoked potential, SSVEP; electromyogram, EMG; extension to multivariate synchronization index, EMSI.

**Figure 6 sensors-21-04578-f006:**
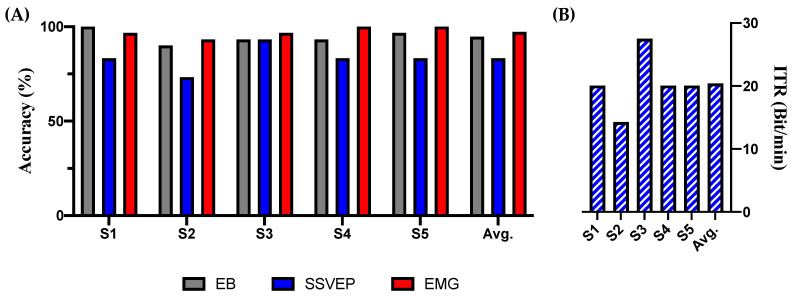
Results of the experiment. Performances of steady state visual evoked potential (SSVEP) were used for three channels and 4 s of data. (**A**) Accuracy of triple eye-blink (EB), SSVEP, and electromyogram (EMG). (**B**) Information transfer rate (ITR) of SSVEP.

**Figure 7 sensors-21-04578-f007:**
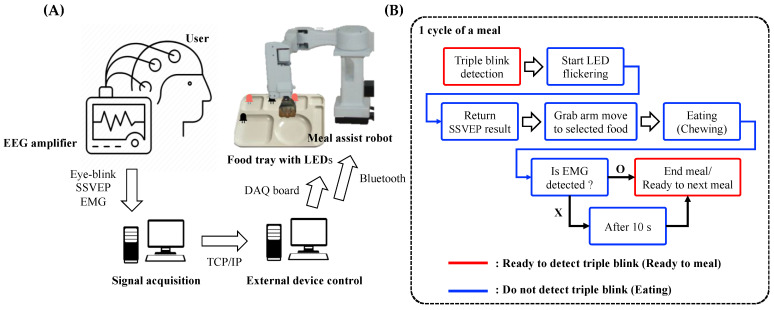
Schematic of hybrid brain–computer interface-based meal-assist robot system. (**A**) Graphical flow chart for used devices. (**B**) Flow chart for actual process including triple blink, Electroencephalogram (EEG) for steady-state evoked potential (SSVEP) and electromyogram (EMG).

**Table 1 sensors-21-04578-t001:** Performance (Accuracy, ITR ^1^) of steady SSVEP ^2^ according to conditions (used number of channels, epoch period).

Condition	SSVEP	S1	S2	S3	S4	S5	Average
1 channel 3 s	Accuracy (%)	60.00	60.00	86.67	73.33	76.67	71.33
	ITR (Bit/min)	11.02	11.02	29.78	19.04	21.43	18.46
3 channels 3 s	Accuracy (%)	60.00	73.33	76.67	70.00	73.33	70.67
	ITR (Bit/min)	11.02	19.04	21.43	16.81	19.04	17.47
1 channel 4 s	Accuracy (%)	66.67	63.33	93.33	83.33	80.00	77.33
	ITR (Bit/min)	11.06	9.61	27.53	20.08	18.00	17.26
3 channels 4 s	Accuracy (%)	83.33	73.33	93.33	83.33	83.33	83.33
	ITR (Bit/min)	20.08	14.28	27.53	20.08	20.08	20.41

^1^ ITR: Information transfer rate. ^2^ SSVEP: Steady state visual evoked potential.

**Table 2 sensors-21-04578-t002:** FPR ^1^ during whole experimental time of EB ^2^ and EMG ^3^.

FPR	S1	S2	S3	S4	S5	Average
EB (times/min)	0.19	0	0	0.19	0.19	0.11
EMG (times/min)	0.19	0	0	0.1	0.1	0.08

^1^ FPR: False positive rate. ^2^ EB: Triple eye-blink. ^3^ EMG: Electromyogram.

## Data Availability

The data presented in this study are openly available in Github at https://github.com/devhaji/HBCI_MAR_Sample (accessed on 29 June 2021).

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
