# Peer review of "A Hybrid Brain–Computer Interface for Real-Life Meal-Assist Robot Control"

_sensors, 2021, doi:10.3390/s21134578_

Round 1
Reviewer 1 Report
This is a nice paper where the author present a hybrid BCI to allow people to eat. While the focus is on the elderly, this could also be envisioned to be used for otherwise younger impaired people with motor disorders. The BCI involved 3 steps on the part of the user, which are: eyeblinks; attending to a LED; chewing. This is repeated so long as necessary. The BCI has separate processing steps for each of these steps.
One major unaddressed point in this work is: How much effort is there on the part of the caregiver, as it currently stands? I'd imaging there is a vast amount of work involved setting up everything: preparing the food, starting up the machine(s), training the subject with the necessary data to (later on) facilitate the on-line eating (was each subject trained invididually?). All of these things to me are the same as "User Friendliness". It's one thing to make a prototype, but another thing entirely to make (other) people use it. See for some more information f.e. James Sulzer's article on rehabilitation devices: https://jneuroengrehab.biomedcentral.com/articles/10.1186/s12984-021-00862-y. The work you present is still relevant though, and in my opinion warrants publication, but I'd rather frame it differently as proof-of-principle work. The actual use and how one would go about it with real patients, is a different question (and study). You could write about this in the discussion.
Another point is the accessibility of data. The section regarding "Data availability Statement" is still filled up with the original placeholder text. It would be great to have a sample of (de-identified) data, and a github repository, with all of the matlab codes, so that this allows others to investigate what is going on and how the data actually looks like.
P2 L56: "The algorithms and hardware of the system were verified by conducting an offline experiment that acquired only EEGs without a meal-assist robot." This sentence could mean that your entire work is validated off-line. This probably isn't the intention? Rather, the off-line part is used to calibrate the EEG (in what way?) to better help the real-time EEG algorithms to figure out what is going on in the real-time part? Because if there is nothing used from the off-line part, there is no reason to do any calibrations and you can just skip right through to the real-time part.
P2 L61: "in which only the signals obtained from the EEG were utilized": this sentence, as written, led me to think that you do not use a hybrid BCI. This is, in fact, true: Only the EEG is used to (1) detect eyeblinks, (2) detect which LED, (3) when chewing stops. I thought Hybrid BCI's combine EEG with other types of (non-EEG) signals?
P3 L88: "Prior to the start of the experiment, the subjects were presented with instructions for the triple eye-blink, observing the indicated LED for SSVEP, and chewing gum for EMG." This is a bit unclear. Does this all happen at the same time? The wordings do not imply a sequence of events, with the EMG always as last. Instead of 'In case','In case','In case', try using 'Initially, 'Secondly/Following',' 'Finally'? Also, you could specifically mention that in your cycles, only the LED is altered. Additionally, you could change your Figure 2 (LED flickering) to illustrate an LED instead of a crosshair - the stimulus on the computer might be a crosshair, but this is not supposed to be attended. Instead of 'Waiting', you could say 'Instruction'? While a BCI is waiting and analyzing, the subject is reading instuctions and blinking, looking at a LED, or chewing.
P4, paragraph 2.1.4: It's not clear to me which kinds of calibrations (your offline session) experiment you use when doing the real-time part (your online session). Which parameters were acquired?
P4 L147: "The proposed system allowed users to eat their own meals completely, unlike conventional systems that require assistance." This is not in fact true. An operator must still set everything up, make the meals, carefully tune the software, etc. Perhaps other systems (which ones?) require specific operator input per session (what kind of input?), while your system requires no trial-based intervention during the course of the meal. In this way you still claim improvement in set-up and procedures over other BCI's, without claiming that no assistance is not required.
P4 L117: What is the reference to FPz? (it's capital P here, too)
P5, Figure 4: Were you affected by the 50Hz artifacts? BCI EEG typically has lots of those, unless you have some active electrode system.
P6, Table 1: It seems that using 3 electrodes seems to perform worse, with the 3 seconds condition. Are these number across all N=5 subjects (include the # of subjects and the trials per subject into the Table/Figure legends). Mention a word in the discussion why this could be?
P6 L178: "(4) 3 channel: O1/ Oz/O2, epoch period: 3 s": did you mean here to say: 4 s epoch period? (note on p.6)
Later on you say 'The SSVEP results of (4) are depicted in Figure 5'. You can make this bit of information into 1 sentence stating at the start of this section: 'SSVEP results (table 4) show that... or something along those lines. Now it detracts from the main information, that 3ch with 4 seconds is the most accurate performing set of parameters.
P6 Fig5: Why present here EB, EMG and then SSVEP, while in your real experiment, the sequence is EB, SSVEP and EMG as last? Better to keep consistency across the paper. You could even harmonize color schemes with Figure 2. Also mention N subjects and M trials into the legend.
P6 Table 2: How are FPR and accuracy (%) for eyeblinks and EMG linked to each other?
P7 L 205: The chewing seems rather vigirous an not what a normal person would do? How does your system fare if chewing EMG is not as pronounced? I am not sure if anyone ever looked at the properties of the "chewing" EMG? There is already some work into the "chewing EMG": https://pubmed.ncbi.nlm.nih.gov/30724370/. Here, it seems to matter what type of food you take, on what kind of EMG you acquire with it. I don't expect a thorough investigation, but you could mention this in the discussion.
This information can be used to improve your EMG FDR. (note on p.8)
P8 L262: "primarily on men": This leaves the possibility that you also had females. You should re-word this to: We conducted this study only on men. Also it's state "were did not vary", and there is one word too many here.
P8 L 283: "based on the SSVEP.": With eye tracking, you can just identify where people are looking at. Could you discuss why to use the SSVEP approach in the discussion? It could be that this is still in prototyping stages as an adjuct or alternative method apart from Eye Tracking, but that might become more important in the future? One might think of approaches where SSVEP types of approaches might be relevant/natural to use, such as f.e. in people with implanted electrodes. There could be ethical dilemmas with that however.
Reviewer 2 Report
I would first like to congratulate and thank the authors Jihyeon Ha , Sang In Park, Chang-Hwan Im and Laehyun Kim for their work.
In the submitted manuscript, they describe a BCI system for control of assisted feeding, an often overlooked application of BCI which has real-world potential and will see increased demand with the worldwide aging population. The described system is simple but effective and its accuracy and ITR are clearly reported. The authors have clearly thought through the many challenges of designing a real-life system which does not place many demands on the user. This is evident from their choices of features and computation. The study is simple but important in contributing to steer the field towards thinking about BCIs in real-life, not in the safe confines of the lab. I recommend its publication, after some minor revisions.
In terms of strong points of the paper:
- Introduction framing is concise and effective
- Choice of simple actions means system should be usable easily by its target audience ie the elderly and disabled persons
- Choice of simple features means system does not require lengthy user training or demanding computation
- The full meal cycle is implemented in a clean and straightforward way which doesn't ask a lot out of the user
The following points could be improved:
- Overall the quality of English used is good, but the common day-to-day usage of the word "meal" refers to full consumption of the food presented. I think in many instances "portion" or "morsel" would be more appropriate to the meaning the authors wish to convey;
- I would like to see a more detailed description of the algorithms used for feature detection, both eye blink and chewing EMG. A schematic with mock waveforms and the consequences of each operation on the original EXG waveform would be very effective. As I'm sure the authors will agree, in BCI systems it's fundamental to understand the inner workings and assumptions of algorithms in order to better understand and prevent failure modes in their real life operation.
General questions to the authors to consider adding in the discussion or elsewhere in the paper:
- Was a "cancel"/"abort" feature considered?
- If applicable, I would be interested in reading about failure modes that were encountered during development, that is, how and when did the system do wrong?
- Potential for an integrated system, where the computation is performed on edge devices/micro-computers which in future could be a part of the robot itself.
- Being a real-life system, it is trying not to place too many constraints on the user, who needs to move in order to operate the system or consume the morsels. Were any problems encountered with motion artefacts and how were they solved?
Reviewer 3 Report
In this manuscript an EEG-based BCI system for meal-assist robot control is presented. The manuscript is well-written with good structure and flow and good presentation. I enjoyed reading it. The objective is clear and so are the experimental protocol and the methodology. The authors provided supplementary material, improving the readability of the article. Below are some comments to the authors:
- Please include a few recent studies from 2021 dealing with daily-life application of BCIs
Zero, E., Bersani, C., & Sacile, R. (2021). Identification of Brain Electrical Activity Related to Head Yaw Rotations. Sensors, 21(10), 3345.
Antoniou, E., Bozios, P., Christou, V., Tzimourta, K. D., Kalafatakis, K., G Tsipouras, M., ... & Tzallas, A. T. (2021). EEG-Based Eye Movement Recognition Using the Brain–Computer Interface and Random Forests. Sensors, 21(7), 2339.
- In the Abstract the authors claim that “Electromyograms were recorded from temporal channels” whereas in the last paragraph of the Introduction the authors claim that “only the signals obtained from the EEG were utilized”. Apparently the authors analyzed the EMG shallowing artifact from temporal channels; however the type of signal is EEG. Please clarify this to avoid any misunderstanding.
- In the title of subsection 2.1.2, Please replace “Paradigm” with “Protocol”
- My only and main concern is the small number of participants that I am sure the authors are aware of. Thus, I suggest the authors to collect more data (perhaps also from women) to validate and improve the quality of the results.

Reviewer 4 Report
Shortcomings:
- “Moreover, we measured the accuracy, information transfer rate, and false positive rate during experiment on five subjects.” This is a fact but provides no more information.
- “we measured the accuracy, information transfer rate, and false positive rate during experiment on five subjects. The average performance over 30 cycles were evaluated, and the results revealed that the proposed system could improve the user-friendliness of such assistance”. Is the second statement related to the previous one? I get such an impression. I was not able to make a conclusion about the improvement of friendliness after reading the manuscript. Where is such evidence?
- “Systems based on augmented/virtual reality (AR/VR) [9], voice [7], and electroencephalograms (EEG) [5,6].”. The statement is without a predicate. Moreover, the statement can be connected to the previous one more tightly.
- “[11-16];”. Specifics must be provided for every reference.
- “Recently, various BCI studies have been conducted using the advantages of each method.”. The statement without references cannot be present.
- “The algorithms and hardware of the system were verified by conducting an offline experiment that acquired only EEGs without a meal-assist robot.”. An offline experiment does not guarantee that the online experiment will go the same way.
- “chewing gum for EMG” is more expressive for the movement of muscles than just consuming a meal.
- “This study only reported offline results for five male subjects.” The online experiment was performed, as well. What is the relation between online and offline experiments? Clarification is needed.
- “The SSVEP-based BCI delivered the highest accuracy among the BCI methods and did not require training”. Reference is needed.
- An offline experiment was performed. Some measurement numbers were delivered. What conclusions can be drawn? Comparison is needed with similar systems.
Round 2
Reviewer 3 Report
The authors performed all the changes and clearly answered all the comments/concerns. The manuscript's quality was very good from the initial submission and only a few changes needed to be performed. From my point of view, the manuscript can now be accepted for publication.